# Examining neighborhood-level hot and cold spots of food insecurity in relation to social vulnerability in Houston, Texas

Ryan Ramphul[1]*, Linda Highfield[2], Shreela Sharma[1]

**1** Department of Epidemiology, Human Genetics & Environmental Sciences, The University of Texas Health Science Center at Houston, Houston, Texas, United States of America, **2** Department of Management, Policy and Community Health, The University of Texas Health Science Center at Houston, Houston, Texas, United States of America

* Ryan.ramphul@uth.tmc.edu

## Abstract

Food insecurity is prevalent and associated with poor health outcomes, but little is known about its geographical nature. The aim of this study is to utilize geospatial modeling of individual-level food insecurity screening data ascertained in health care settings to test for neighborhood hot and cold spots of food insecurity in a large metropolitan area, and then compare these hot spot neighborhoods to cold spot neighborhoods in terms of the CDC's Social Vulnerability Index. In this cross-sectional secondary data analysis, we geocoded the home addresses of 6,749 unique participants screened for food insecurity at health care locations participating in CMS's Accountable Health Communities (AHC) Model, as implemented in Houston, TX. Next, we created census-tract level incidence profiles of positive food insecurity screens per 1,000 people. We used Anselin's Local Moran's I statistic to test for statistically significant census tract-level hot/cold spots of food insecurity. Finally, we utilized a Mann-Whitney-U test to compare hot spot tracts to cold spot tracts in relation to the CDC's Social Vulnerability Index. We found that hot spot tracts had higher overall social vulnerability index scores (P <0.001), higher subdomain scores, and higher percentages of individual variables like poverty (P <0.001), unemployment (P <0.001), limited English proficiency (P <0.001), and more. The combination of robust food insecurity screening data, geospatial modeling, and the CDC's Social Vulnerability Index offers a solid method to understand neighborhood food insecurity.

## Introduction

The United States Department of Agriculture (USDA) defines food insecurity as a household-level economic and social condition of limited or uncertain access to adequate food [1]. They further divide food insecurity into two levels, with "low food security" referring to reduced quality, variety, or desirability of diet, but no reduction of food intake, and "very low food security," referring to disrupted eating patterns and reduced food intake [1]. In 2020, an estimated 13.8 million households in the US reported food insecurity at some time during the

**Data Availability Statement:** The data cannot be shared publicly because of ethical restrictions and data protection issues, as our dataset includes potentially identifying or sensitive patient information. Data are available from the UTHealth

Houston Committee for the Protection of Human Subjects (contact via cphs@uth.tmc.edu) for researchers who meet the criteria for access to confidential data.

**Funding:** The author(s) received no specific funding for this work.

**Competing interests:** The authors have declared that no competing interests exist.

year [1]. There were an estimated 724,750 food insecure individuals in Greater Houston in 2018, and the estimated food insecurity rate was about 16.6%, which was roughly 4 percentage points above the national average in that year [2].

In a systematic review of literature on the impact of food insecurity on health outcomes, Gunderson and Ziliak [3] examine recent research evidence of the health consequences of food insecurity for children, non-senior adults, and seniors throughout the US. Regarding children, they found studies suggesting that food insecurity is associated with an increased risk of birth defects, anemia, lower nutrient intake, cognitive problems, aggression, and anxiety [3]. They also found research indicating that food insecurity is associated with higher risks of children being hospitalized, having asthma, behavioral problems, depression, suicide ideation, poor oral health, and poor overall health [3]. Regarding non-senior adults, they cite studies showing that food insecurity is associated with decreased nutrient intake, increased rates of mental health problems, diabetes, hypertension and hyperlipidemia, poor sleep outcomes, and poor overall health [3]. Finally, they point to studies indicating that food-insecure seniors are more likely to be in poor health, depressed, and have limited daily activities compared to their food-secure peers [3].

The aim of this study is to utilize geospatial modeling of individual-level food insecurity screening data ascertained in health care settings to test for neighborhood hot and cold spots of food insecurity in a large metropolitan area, and then compare these hot spot neighborhoods to cold spot neighborhoods in terms of the CDC's Social Vulnerability Index and subdomains. Testing for neighborhood-level hot/cold spots of food insecurity is necessary because little is known about the geographical nature of food insecurity. While other social determinants of health like poverty and education levels are tracked extensively at the neighborhood level through surveillance mechanisms like the Census's American Community Survey, food insecurity isn't. With a greater understanding of the neighborhood-level factors that affect area-level food insecurity, health care providers, public health practitioners and policy makers can potentially craft interventions aimed at mitigating it.

In 2008, the CDC produced the Modified Retail Food Environment Index (mRFEI), which indicated the percentage of healthy food retailers by census tract. The USDA later demarcate "food deserts," or areas that lack stores that sell healthy and affordable food, by introducing low income, low access (LILA) census tracts, where low income is defined by poverty rates and median family income, and low access by proximity to supermarkets or large grocery stores [4]. LILA and mREFI census tracts are widely used in food environment studies but assume that proximity to food retailers, or the presence of food retailers in a geographical area, is related to the ability to access food. Studies show, however, that food choices and purchases are not necessarily associated with proximity to food outlets but are likely a function of personal preferences, cultural factors, social norms, and the ability to afford foods [5–8]. Additional research validates this concept, showing that the introduction of low-cost grocery stores into underserved "food desert" neighborhoods does not significantly impact shopping behaviors, household food availability, or the health of residents [9–12].

Feeding America, a nationwide non-profit organization aimed at mitigating hunger and food insecurity, offers one of the few methods of identifying neighborhood food insecurity that doesn't involve proximity to food outlets, in its Map the Meal Gap tool. Defining food insecurity as the lack of reliable access to a sufficient quantity of affordable, nutritious food, the Mapping the Meal Gap tool estimates regional and area-level food insecurity prevalence using publicly available data on neighborhood unemployment, poverty, and other household characteristics [13]. To our knowledge, however, no studies have explored the use of address-level food insecurity screening data from healthcare settings, geospatial modeling, and the CDC's Social Vulnerability Index, to better understand neighborhood food insecurity.

## Materials and methods

### Data

This is a cross-sectional secondary data analysis of food insecurity screening data collected from Medicare and Medicaid beneficiaries who participated in the Centers for Medicare & Medicaid Service's (CMS) Accountable Health Communities (AHC) Model, which was piloted by the University of Texas Health Science Center at Houston School of Public Health. According to CMS, the AHC Model seeks to address gaps between clinical care and community services by testing whether identifying and addressing health-related social needs through screening, referral and community navigation, impacts health care costs and reduces health care utilization [14]. Screening was offered to all community-dwelling Medicare and Medicaid beneficiaries at several clinical delivery sites across Greater Houston. These sites include five hospitals affiliated with two large Texas Medical Center (TMC) health systems, and four outpatient clinics affiliated with one of the area's largest ambulatory groups.

### Food insecurity

Like many models that screen for food insecurity, The AHC model adopted The Hunger Vital Sign[TM] screening tool [15]. Hunger Vital Signs is a two-question screening tool, suitable for clinical or community outreach use, that identifies risk for food insecurity if families answer that either or both of the following statements are "often true" or "sometimes true," versus "never true":

- "Within the past 12 months we worried whether our food would run out before we got money to buy more".

- "Within the past 12 months the food we bought just didn't last and we didn't have money to get more".

Children's Health Watch, a nonpartisan network of pediatricians, public health researchers, and policy and child health experts validated the Hunger Vital Sign[TM] tool with a sample of 30,000 caregivers. They found excellent sensitivity (97%) and specificity (83%) with the Hunger Vital Signs tool compared to the much longer US Household Food Security Scale (HFSS) screening tool, which is considered the "gold standard" in assessment and identification of food security [15].

### Statistical analysis

After obtaining IRB approval from the University of Texas Health Science Center at Houston's Committee for the Protection of Human Subjects, we pulled a total of 7,658 food insecurity screening records, collected at AHC sites in Harris County, from August 2018–January 2020. Participants' home addresses were geocoded using ArcGIS Pro Street Map Premium. Partial addresses, unrecognizable addresses, PO Box addresses, and addresses that fall outside of Harris County were excluded, providing a sample size of 6,749 useable addresses that geocoded successfully to a street address. 3,636 participants screened positive for food insecurity and 3,113 screened negative. After addresses were mapped using ArcGIS, we created incidence profiles of people who screened positive for food insecurity per 1,000 people, by census tract, and analyzed the data spatially.

We utilized Anselin's Local Moran's I statistic to identify statistically significant clusters of census tracts with a high/low incidence of positive food insecurity screens. Anselin's Local Moran's I statistic, also known as the Local Indicators of Spatial Association (LISA) statistic, does this by creating a neighborhood around each census tract and calculating a Moran's I

score for each neighborhood, which compares the neighborhood to the study area [16]. If the Moran's I score is positive then that census tract has similar values to its neighbors and is part of a suspected cluster, if it's negative, it has dissimilar values from its neighborhood and is part of a suspected outlier [16]. We utilized the queen's contiguity method, a commonly used method to model spatial relationships in public health analysis, to determine the neighborhood around each census tract, which means that census tracts that share sides or vertices with a given census tract are part of its neighborhood [16].

To test for statistical significance, the Local Moran's I statistic randomly takes values from other census tracts in the study area, imputes them into a given neighborhood and recalculates the Moran's I score for that neighborhood [16]. It does this 9,999 times as part of a Monte Carlo simulation aimed at creating a reference distribution to compare the observed local Moran's I score with one created by random permutations [16]. If less than 5% of the Local Moran's I values generated from permutations display more clustering than the original data, then the data displays significant clustering [16]. Clusters of high values are known as hot spots and clusters of low values are known as cold spots [16].

In the context of this research, hot spot census tracts are census tracts with high incidences of positive food insecurity screens, surrounded by neighboring census tracts with high incidences of food insecurity screens, tested for statistical significance. Cold spots are census tracts with low incidences of positive food insecurity screens, surrounded by neighboring census tracts with low incidences of positive food insecurity screens, tested for statistical significance. Analyses were performed using the Cluster and Outlier Analysis tool in ArcGIS Pro Version 2.9.2.

Next, we compared hot spot and cold spot census tracts in terms of neighborhood characteristics, using the CDC's Social Vulnerability Index (SVI). The CDC's Social Vulnerability index uses 15 census tract-level variables to create subdomain scores for socioeconomic status, household composition & disability, minority status & language, and housing & transportation. Scores for each subdomain are then used to create an overall social vulnerability index score. For subdomain scores and overall scores, values close to 0 indicate low vulnerability and values close to 1 indicate high vulnerability. Since all of the variables we looked at were non-parametric, we utilized a Mann Whitney-U test, performed in SPSS Version 28.0.1.1, to compare median rank values in hot spot census tracts versus cold spot census tracts for 15 census variables, all four SVI subdomain scores, and overall SVI scores.

## Results and discussion

### Results

We Identified 66 statistically significant hot spot census tracts of food insecurity incidence and 150 cold spots (Table 1 and Fig 1). The rest of the census tracts in Harris County (570) were either not part of any statistically significant clusters or were outliers. The Mann-Whitney U

**Table 1. Cluster and outlier analysis of food insecurity incidence, by census tract.**

|  | Harris County Census Tracts (N = 786) | Significance[a] |
|---|---|---|
| Hot Spots (High-High Clusters) | 66 (8.4%) | < .05 |
| Cold Spots (Low-Low Clusters) | 150 (19%) | < .05 |
| Hot Outliers (High-Low Clusters) | 1 (0.1%) | < .05 |
| Cold Outliers (Low-High Clusters) | 4 (0.5%) | < .05 |
| Not Significant | 565 (71.8%) |  |

[a]Anselin's Local Moran's I Significance Level (.05).

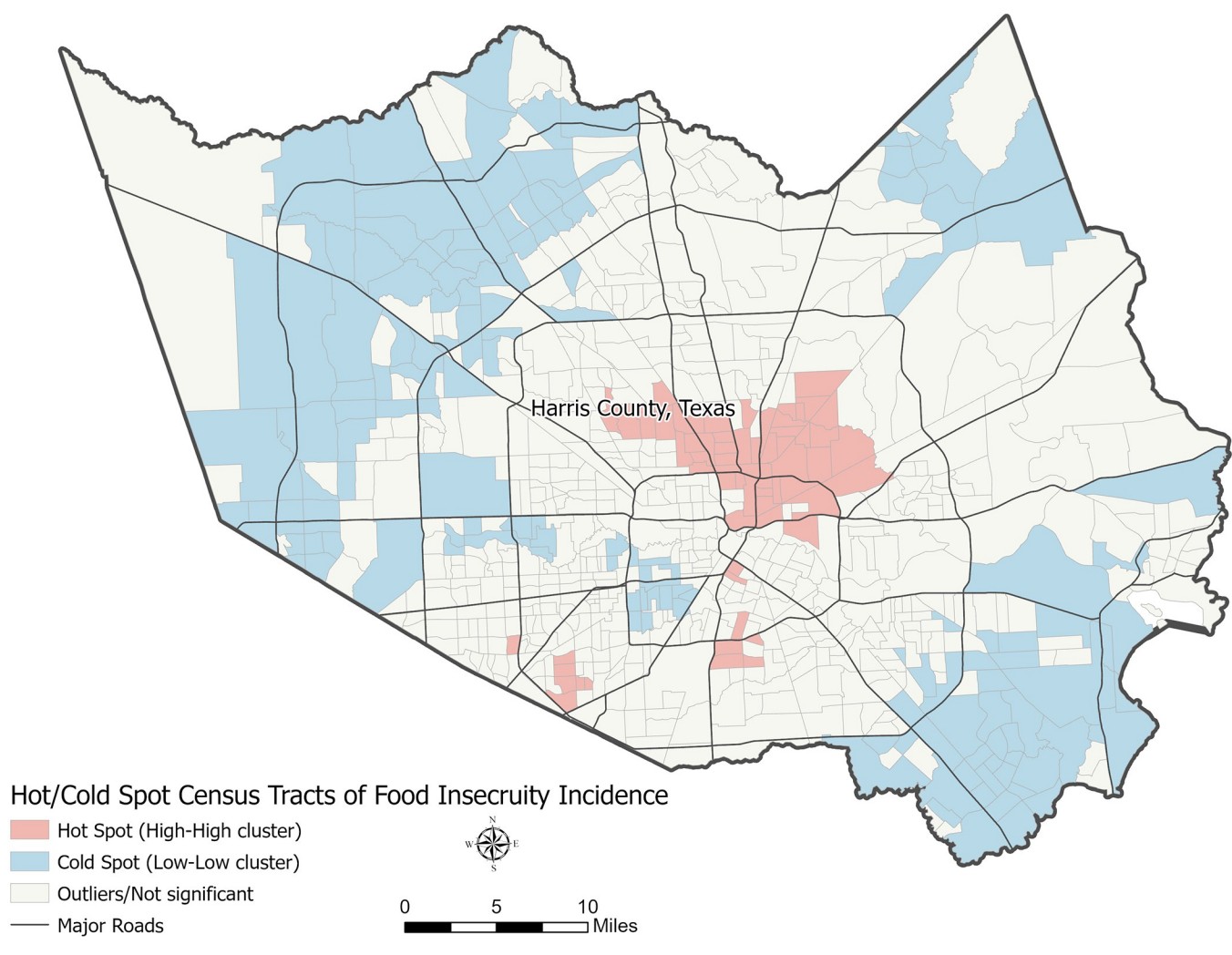

**Fig 1. Hot/Cold spot map of food insecurity incidence, by tract.**

test indicated that hot spot census tracts were significantly more socially vulnerable than cold spot census tracts, as measured by SVI subdomain scores and overall scores (Table 2). In terms of SVI domain 1, socioeconomic status, hot spot census tracts had higher percentages of people below the federal poverty level, unemployed people, people with no high school diploma, and lower per capita incomes.

Regarding domain 2, household composition & disability, they had statistically higher percentages of disabled people and single parent households. In SVI domain 3, minority status and language, hot spot census tracts had statistically higher percentages of minorities and people with limited English proficiency than cold spot tracts. For domain 4, housing and transportation, hot spot census tracts of food insecurity had higher percentages of household crowding, no vehicle ownership, and group quarters than cold spot tracts. We created a choropleth map of overall social vulnerability, by census tract, for visual comparison (Fig 2).

## Discussion

This study utilized geospatial modeling of individual-level food insecurity screening data ascertained in health care settings to identify neighborhood hot and cold spots of food

**Table 2. Comparison of hot/cold spot tracts of food insecurity incidence, in terms of the CDC's Social Vulnerability Index, subdomains, and individual components.**

| | Tract-Level Characteristic | Hot Spot Tracts Median (Mean Rank) | Cold Spot Tracts Median (Mean Rank) | Test Statistic | p-value[a] |
|---|---|---|---|---|---|
| Domain 1: Socioeconomic status | % Below Poverty | 30.75% (171.74) | 6.05% (80.67) | 776.00 | < .001 |
| | % Unemployed | 7.20% (139.67) | 4.00% (94.78) | 2892.50 | < .001 |
| | Per capita income | $16,708 (48.20) | $39,429 (135.03) | 970.00 | < .001 |
| | % With no HS diploma | 30.10% (169.92) | 5.30% (81.48) | 896.50 | < .001 |
| | Domain score | .84 (169.83) | .19 (81.51) | 902.00 | < .001 |
| Domain 2: Household Composition & Disability | % Age 65+ | 11.75% (114.16) | 11.35% (106.01) | 4576.50 | .377 |
| | % Age 17 or younger | 26.10% (104.58) | 25.90% (110.22) | 4691.50 | .541 |
| | % Disabled | 14.15% (164.94) | 7.5% (83.67) | 1225.00 | < .001 |
| | % Single-parent households | 13.85% (147.35) | 6.85% (91.41) | 2386.00 | < .001 |
| | Domain score | .69 (165.26) | .22 (83.53) | 1204.00 | < .001 |
| Domain 3: Minority Status & Language | % Minority | 96.10% (175.04) | 47.00% (79.22) | 558.50 | < .001 |
| | % Speak English "Less than well" | 13.35% (154.52) | 3.30% (88.25) | 1912.50 | < .001 |
| | Domain score | .87 (169.06) | .41 (81.85) | 953.00 | < .001 |
| Domain 4: Housing & Transportation | % Multi-unit structures | 12.90% (105.81) | 17.80% (109.68) | 4772.50 | .675 |
| | % Mobile homes | .40% (113.08) | .00% (106.49) | 4648.00 | .432 |
| | % Crowding | 6.10% (152.88) | 1.95% (88.97) | 2021.00 | < .001 |
| | % No vehicle | 13.90% (172.46) | 2.15% (0.36) | 728.50 | < .001 |
| | % Group quarters | .30% (136.72) | .00% (96.08) | 3087.50 | < .001 |
| | Domain score | .71 (163.33) | .26 (84.37) | 1331.00 | < .001 |
| Overall | Overall SVI score | .89 (172.20) | .20 (80.47) | 745.00 | < .001 |

[a]Mann Whitney-U Test Significance level (0.05).

insecurity in a large metropolitan area. This adds to the knowledge base on neighborhood food insecurity by providing evidence that food insecurity exhibits a neighborhood clustering pattern versus a null hypothesis of being randomly distributed throughout the region. This is noteworthy because little is understood about the geographical nature of food insecurity, due to an absence of robust surveillance on the issue. Researchers used to postulate that food-insecure people resided in "food deserts" until a growing body of evidence suggested otherwise [9–12]. Many food insecurity researchers and advocates now estimate neighborhood food

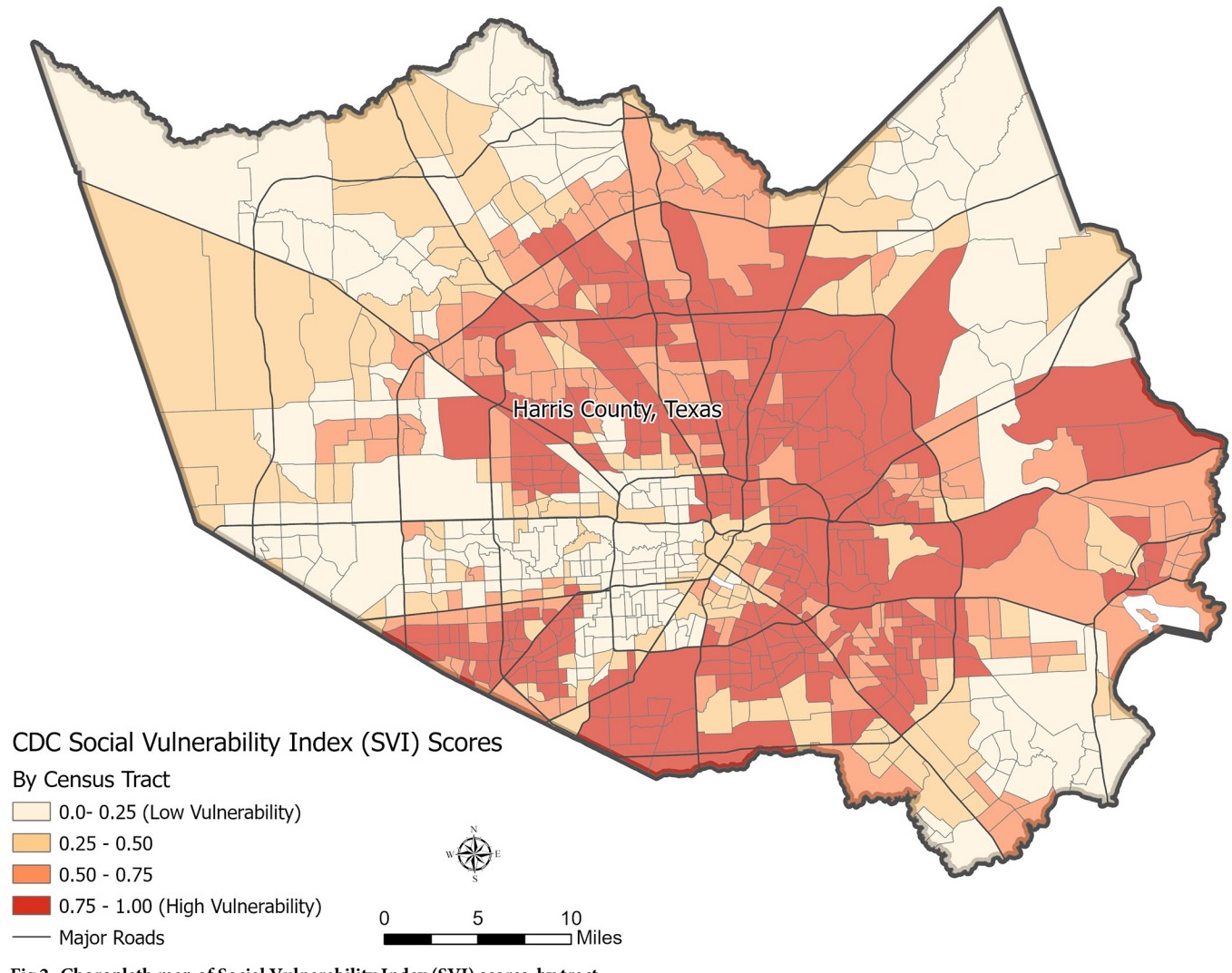

**Fig 2. Choropleth map of Social Vulnerability Index (SVI) scores, by tract.**

insecurity rates using extant data on poverty and unemployment, but this analysis uses more precision inputs and more robust modeling.

This analysis also provides evidence that neighborhood hot spots of food insecurity have higher levels of social vulnerability than cold spots, as measured by the CDC's Social Vulnerability Index, it's subdomains, and individual census-tract level variables. While the relationship between social vulnerability and food insecurity is well established in scientific literature, this analysis is unique because it is the first analysis, to our knowledge, to examine this issue using geospatial modeling and the CDC's Social Vulnerability Index, a well-established and thorough measure of neighborhood vulnerability. This implies that even if public health practitioners and food security advocates do not know where food insecurity is more or less prevalent in a region, they can target their efforts to high SVI communities and likely reach food insecure populations.

## Strengths and limitations

### Strengths

We are unaware of any analysis of a universal offer to screen approach for food insecurity, such as the one used in the AHC Model, where every Medicaid or Medicare patient seen at

participating clinics is offered screening. We are also unaware of any other research utilizing geospatial modeling to identify census tract-level hot/cold spots of food insecurity in a large metropolitan area–a methodology that can guide the deployment of neighborhood-level interventions in a more precise manner. Kolak, Abraham, & Talen [17] point out in their analysis of type 2 diabetes clusters in a primary care population in Chicago, that adapting spatial methods to smaller, more localized populations, may provide additional strategies for identifying localized areas of health risk for targeting interventions and improving care in smaller panel populations.

## Limitations

The primary limitation in this analysis is the sample size of participants screened for food insecurity. Ideally, there would be several more data collection sites and a much larger sample of screens. Additionally, the AHC Model only screens people enrolled in Medicaid and Medicare, excluding people who have private insurance or are uninsured or do not access the health care system. This limits the sample and adds selection bias. Further, the data sample we used in this analysis was collected before the onset of the COVID-19 pandemic, which significantly increased food insecurity despite emergency legislation that put more resources into food assistance programs, increased unemployment benefits, and provided stimulus payments [18]. A final limitation of this study is that it only examines food insecurity in binary terms, instead of considering the different levels of food insecurity.

## Conclusions

In the absence of more robust surveillance of food insecurity, using data from people screened for food insecurity in health care settings, geospatial modeling, and the CDC's Social Vulnerability Index, offers a solid approach for understanding food insecurity at the neighborhood level. As the practice of screening for food insecurity in health care settings continues to grow, future research should examine this methodology with a larger sample of food insecurity screens, over several regions, or even throughout the entire state. Future research should also utilize this method to analyze neighborhood food insecurity in the years after the onset of the COVID-19 pandemic.

## Acknowledgments

Conceived of the idea, performed the computations, and took the lead on writing: RR.
Verified the analytical methods and supervised the findings: LH,FLR,SS.

## Author Contributions

**Investigation:** Ryan Ramphul.

**Methodology:** Ryan Ramphul.

**Project administration:** Ryan Ramphul.

**Supervision:** Ryan Ramphul, Linda Highfield, Shreela Sharma.

**Visualization:** Ryan Ramphul.

**Writing – review & editing:** Ryan Ramphul.

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
