## [Decision Letter · Decision Letter 0]

27 Aug 2022

PONE-D-22-11199Examining Neighborhood-Level Hot and Cold Spots of Food Insecurity In Relation to Social Vulnerability in Houston, Texas

PLOS ONE

Dear Dr. Ramphul,

Thank you for submitting your manuscript to PLOS ONE. After careful consideration, we feel that it has merit but does not fully meet PLOS ONE’s publication criteria as it currently stands. Therefore, we invite you to submit a revised version of the manuscript that addresses the points raised during the review process.

The manuscript has been evaluated by four reviewers, and their comments are available below.

The reviewers have raised a number of  concerns regarding the methodology and reporting of this study.

Could you please revise the manuscript to carefully address the concerns raised?

We look forward to receiving your revised manuscript.

Kind regards,

Johannes Stortz

Staff Editor

PLOS ONE

Journal Requirements:

2. We note that Figure 1 & 2 in your submission contain [map/satellite] images which may be copyrighted. All PLOS content is published under the Creative Commons Attribution License (CC BY 4.0), which means that the manuscript, images, and Supporting Information files will be freely available online, and any third party is permitted to access, download, copy, distribute, and use these materials in any way, even commercially, with proper attribution. For these reasons, we cannot publish previously copyrighted maps or satellite images created using proprietary data, such as Google software (Google Maps, Street View, and Earth). For more information, see our copyright guidelines: http://journals.plos.org/plosone/s/licenses-and-copyright.

1. You may seek permission from the original copyright holder of Figure 1 & 2 to publish the content specifically under the CC BY 4.0 license.  

Reviewers' comments:

Reviewer's Responses to Questions

**Comments to the Author**

1. Is the manuscript technically sound, and do the data support the conclusions?

Reviewer #1: Yes

Reviewer #2: Partly

Reviewer #3: Partly

Reviewer #4: Yes

2. Has the statistical analysis been performed appropriately and rigorously? 

Reviewer #1: Yes

Reviewer #2: I Don't Know

Reviewer #3: Yes

Reviewer #4: Yes

3. Have the authors made all data underlying the findings in their manuscript fully available?

Reviewer #1: Yes

Reviewer #2: Yes

Reviewer #3: No

Reviewer #4: No

4. Is the manuscript presented in an intelligible fashion and written in standard English?

Reviewer #1: Yes

Reviewer #2: Yes

Reviewer #3: Yes

Reviewer #4: Yes

5. Review Comments to the Author

Reviewer #1: To know about neighborhood food insecurity authors geocoded food insecurity screening data provided by health provider, along with geospatial modeling, and Social Vulnerability Index. They found that hot spot tracts were having higher overall social vulnerability index scores, greater subdomain scores, and increased poverty, unemployment, limited English proficiency. Overall this is an interesting study. I think it is well written and properly explained but few minor suggestions are there to improve the paper:

When you say plethora of research …. and see last line in the para references provided same and only 1 in the whole paragraph. Better cite more literature here. See below

“A plethora of research suggests that food insecurity is not only highly prevalent but associated with several poor health outcomes. Regarding children, studies suggest that food insecurity is associated with an increased risk of birth defects, anemia, lower nutrient intake, cognitive problems, aggression, and anxiety (1). Research also indicates that food insecurity is associated with higher risks of children being hospitalized, having asthma, behavioral problems, depression, suicide ideation, poor oral health, and poor overall health (1). Regarding non-senior adults, studies show that food insecurity is associated with decreased nutrient intake, increased rates of mental health problems, diabetes, hypertension and hyperlipidemia, poor sleep outcomes, and poor overall health (1). Finally, studies indicate that food insecure seniors are more likely to be in poor health, depressed, and have limited daily activities compared to their food-secure peers (1).

In introduction I think extra details on food security outcomes are given than needed. Instead of three paragraphs on outcomes better be concise.

Last two Paragraphs of discussion section is about study limitation and strength, So better give a heading “The limitation and Strength.”

Literature and references are not sufficient give some more studies and literature.

Reference 20 is in the conclusion. Better put these lines in discussion/analysis section.

Reviewer #2: Thank you for allowing me to review this interesting manuscript called "Examining Neighborhood-Level Hot and Cold Spots of Food Insecurity In Relation to Social Vulnerability in Houston, Texas." It aimed to assess residential spatial patterns of Medicare and Medicaid beneficiaries who screened positive for food insecurity at health care locations in Harris County, Texas. Even though the study could contribute to the field, some aspects do not allow me to accept it in its current form.

1. The definition of food insecurity in the introduction lacks other essential elements such as food availability. Please, add the reference for the definition used.

2. I am not sure what the need for this study is. This should be stated clearly in the introduction.

3. Please, clarify the aim of the study in the abstract and the text.

4. The food insecurity information was collected during the pandemic. There is extensive evidence that food insecurity increased during the pandemic due to socioeconomic and health problems. How could this affect the results of this research? Is this a limitation of the study?

5. There are different levels of food insecurity. Why were they not considered in the analysis?

6. I suggest adding the definition of hot and cold spots in the methods section and removing it from the results.

7. The aim of the study should guide the conclusion. However, the authors conclude with ideas related to the healthcare system.

8. In the discussion section, the authors state: "Researchers used to postulate that food-insecure people resided in "food deserts" until a growing body of evidence suggested otherwise." Here, it is essential to re-review the definition of food insecurity because it includes the lack of economic and physical access to food.

9. Please, review the names of all tables and figures.

10. I suggest separating the results and discussion sections.

Reviewer #3: This was an interesting article and thank you for the opportunity to review this article. I have suggested a few revisions that could further improve this paper.

List of comments/suggestions:

Abstract: Please include the sample size. If word count permits, consider providing the p values for the significant differences mentioned in the abstract.

Introduction: A citation should be provided for the statement “Food insecurity is highly prevalent in the US”. If possible, provide the most recent statistics on food insecurity prevalence. Consider defining food desert area when it is first mentioned on page 3. In the introduction the importance of specifically using the address-level food insecurity screening data and the spatial clustering methods.

Materials and methods: This section could be clearer if you could break it into subsections under materials and methods. Provide a citation for the Hunger Vital SignTM screening tool. Consider using a flow diagram to show the participants' inclusion/exclusion. There is a typo on page 7, line 3. A citation is needed for the queen’s contiguity method. The second paragraph on page 7, requires a citation. Define the abbreviation SVI on page 8, the first line.

Results and discussion:

I believe you have mistakenly used “… in terms of HPV vaccine uptake” in the title of Table 2. Also, there is a typo in this title. Include the Mann-Whitney U test statistics with the p-values in the table.

I think it will be better if you could add a footnote to the Table 2 indicating what this p-value is. It is in the text but missing from the table.

Conclusion: I feel that except for the final sentence other statements in this section could not be considered as conclusions based on this study's findings. Please include only the conclusions that are based on your findings.

Reviewer #4: This paper provided a unique way to measure neighborhood food insecurity. There are some minor suggestions that would improve this paper:

Results and Discussion

• Stating that finding the prevalence of food insecurity to be 54% in this study compared to Greater Houston at 16.6% is due to comfort with answering food insecurity screening in this setting might be an overstatement. People that attend this clinic were on Medicaid or Medicare, suggesting a higher probability of lower/fixed income.

• Additionally, the method used to measure food insecurity in this study was the Hunger Vital Sign and the full USDA questionnaire may have been the method used to get the Greater Houston prevalence of food insecurity. Authors should report if this may have contributed to a difference in measurement for comparison.

• The title for Table 2 does not make sense and appears to be copied from a different study.

6. PLOS authors have the option to publish the peer review history of their article (what does this mean?). If published, this will include your full peer review and any attached files.

Reviewer #1: No

Reviewer #2: No

Reviewer #3: No

Reviewer #4: No

---

## [Author Response · Author response to Decision Letter 0]

31 Oct 2022

Dear Reviewers:

Thank you for your comments on my manuscript entitled “Examining Neighborhood-Level Hot/Cold Spots of Food Insecurity in Relation to Social Vulnerability in Houston, Texas.” I have edited the manuscript to address each point in the manner outlined below.

Editor: 

• I verified that my manuscript meets PLOS ONE’s style requirements, including those for file naming.

• I removed any copywritten information from Figures 1 and 2.

• I also reviewed my reference list to ensure completeness and correctness.

Reviewer #1: To know about neighborhood food insecurity authors geocoded food insecurity screening data provided by health provider, along with geospatial modeling, and Social Vulnerability Index. They found that hot spot tracts were having higher overall social vulnerability index scores, greater subdomain scores, and increased poverty, unemployment, limited English proficiency. Overall this is an interesting study. I think it is well written and properly explained but few minor suggestions are there to improve the paper:

When you say plethora of research …. and see last line in the para references provided same and only 1 in the whole paragraph. Better cite more literature here. See below

“A plethora of research suggests that food insecurity is not only highly prevalent but associated with several poor health outcomes. Regarding children, studies suggest that food insecurity is associated with an increased risk of birth defects, anemia, lower nutrient intake, cognitive problems, aggression, and anxiety (1). Research also indicates that food insecurity is associated with higher risks of children being hospitalized, having asthma, behavioral problems, depression, suicide ideation, poor oral health, and poor overall health (1). Regarding non-senior adults, studies show that food insecurity is associated with decreased nutrient intake, increased rates of mental health problems, diabetes, hypertension and hyperlipidemia, poor sleep outcomes, and poor overall health (1). Finally, studies indicate that food insecure seniors are more likely to be in poor health, depressed, and have limited daily activities compared to their food-secure peers (1).

• Thank you for this comment. I rewrote this paragraph to clarify that this information came from a systematic review of literature on the impact of food insecurity on health outcomes. 

In introduction I think extra details on food security outcomes are given than needed. Instead of three paragraphs on outcomes better be concise.

• Thank you for this comment. I agree, I removed one of the paragraphs on the effects of food insecurity on health outcomes.

Last two Paragraphs of discussion section is about study limitation and strength, so better give a heading “The limitation and Strength.”

• Thank you for this comment. I added a “Strengths and Limitations” heading.

Literature and references are not sufficient give some more studies and literature.

• Thank you for this comment. In rewriting the manuscript based on the various comments provided, I ended up adding some references but removing others.

Reference 20 is in the conclusion. Better put these lines in discussion/analysis section.

• Thank you for this comment, I rearranged the discussion and conclusion sections to make them flow better. Also, I moved reference 20 from the conclusion section.

Reviewer #2: Thank you for allowing me to review this interesting manuscript called "Examining Neighborhood-Level Hot and Cold Spots of Food Insecurity In Relation to Social Vulnerability in Houston, Texas." It aimed to assess residential spatial patterns of Medicare and Medicaid beneficiaries who screened positive for food insecurity at health care locations in Harris County, Texas. Even though the study could contribute to the field, some aspects do not allow me to accept it in its current form.

The definition of food insecurity in the introduction lacks other essential elements such as food availability. Please, add the reference for the definition used.

• Thank you for this comment. Instead of paraphrasing the definition of food insecurity I updated the text with the USDA’s full definition of food insecurity, complete with information on the levels of food insecurity that they define. 

2. I am not sure what the need for this study is. This should be stated clearly in the introduction.

• This is a good point. I revamped the introduction section of this paper and added a specific paragraph on aim of this study and why it is needed. See paragraph 3.

3. Please, clarify the aim of the study in the abstract and the text.

• This is a good point. I revamped the introduction section of this paper and added a specific paragraph on aim of this study and why it is needed. See paragraph 3.

4. The food insecurity information was collected during the pandemic. There is extensive evidence that food insecurity increased during the pandemic due to socioeconomic and health problems. How could this affect the results of this research? Is this a limitation of the study?

• This is an excellent point. Our food insecurity screening data was collected just before the onset of the pandemic, which ended up exacerbating food insecurity significantly. I added this point to the limitations section of this paper.

5. There are different levels of food insecurity. Why were they not considered in the analysis?

• This is an excellent point. I added some information in the Introduction section about how the USDA’s definition of food insecurity involves different levels of food insecurity. The reason our analysis does not break food insecurity into different levels is because we utilize the Hunger Vital Signs screening tool to screen for food insecurity. The Hunger Vital Signs tool is an adaption of the longer USDA U.S. Household Food Security Survey, that has been validated for effectiveness. Health care providers often use the Hunger Vital Signs tool because it is much easier to administer than the U.S. Household Food Security Survey, but unfortunately the Hunger Vital Signs tool only indicates if someone is food insecure or not. It doesn’t indicate if they are very food insecure.

6. I suggest adding the definition of hot and cold spots in the methods section and removing it from the results.

• Agreed, I added the definition of hot and cold spots to the methods section and removed it from the results.

7. The aim of the study should guide the conclusion. However, the authors conclude with ideas related to the healthcare system.

• This is a good point. I updated the Introduction section with a specific paragraph about the aim of the study and why it’s needed. I also rewrote the conclusion section to complement the aim paragraph, removing much of the text about the healthcare system.

8. In the discussion section, the authors state: "Researchers used to postulate that food-insecure people resided in "food deserts" until a growing body of evidence suggested otherwise." Here, it is essential to re-review the definition of food insecurity because it includes the lack of economic and physical access to food.

• Thank you for this point. I updated the Introduction section with the USDA’s full definition of food insecurity, complete with information on the levels of food insecurity that they define. I also updated paragraph 4 of the Introduction which discusses definitions of food deserts.

9. Please, review the names of all tables and figures.

• Thank you, I reviewed the names of all tables and figures and adjusted accordingly.

10. I suggest separating the results and discussion sections.

• Thank you, I separated the results and discussion sections.

Reviewer #3: This was an interesting article and thank you for the opportunity to review this article. I have suggested a few revisions that could further improve this paper.

List of comments/suggestions:

Abstract: Please include the sample size. If word count permits, consider providing the p values for the significant differences mentioned in the abstract.

• Thank you for this comment, I added sample size and p-values to the abstract.

Introduction: A citation should be provided for the statement “Food insecurity is highly prevalent in the US”. If possible, provide the most recent statistics on food insecurity prevalence. Consider defining food desert area when it is first mentioned on page 3. In the introduction the importance of specifically using the address-level food insecurity screening data and the spatial clustering methods.

• Thank you for these comments. I updated the statement about food insecurity prevalence, and added some updated statistics on the estimated number of households reporting food insecurity. I also added a paraphrased definition of “food deserts” when I first introduce the concept in the paragraph indicated. In addition, I rephrased the introduction section to stress the importance of using address-level food insecurity screening data and spatial clustering methods.

Materials and methods: This section could be clearer if you could break it into subsections under materials and methods. Provide a citation for the Hunger Vital SignTM screening tool. Consider using a flow diagram to show the participants' inclusion/exclusion. There is a typo on page 7, line 3. A citation is needed for the queen’s contiguity method. The second paragraph on page 7, requires a citation. Define the abbreviation SVI on page 8, the first line.

• Thank you for these comments, I separated out the Materials and Methods sections into two separate sections. I also added the citation for the Hunger Vital Sign Screening tool, and updated the typo (changed “than” to “then”) on page 7. In addition, I added a citation for the sentence about the queen’s contiguity method and added a citation for the second paragraph on page 7. Finally, I added text to indicate that SVI refers to the Social Vulnerability index in the first sentence of this paragraph.

Results and discussion:

I believe you have mistakenly used “… in terms of HPV vaccine uptake” in the title of Table 2. Also, there is a typo in this title. Include the Mann-Whitney U test statistics with the p-values in the table.

I think it will be better if you could add a footnote to the Table 2 indicating what this p-value is. It is in the text but missing from the table.

• Thank you for this comment. I corrected the title of the table, added median values and the test statistics, and also added a footnote indicating significance levels. 

Conclusion: I feel that except for the final sentence other statements in this section could not be considered as conclusions based on this study's findings. Please include only the conclusions that are based on your findings.

• This is a good point. I rewrote the conclusion to focus on conclusions based on this study’s findings.

Reviewer #4: This paper provided a unique way to measure neighborhood food insecurity. There are some minor suggestions that would improve this paper:

Results and Discussion

Stating that finding the prevalence of food insecurity to be 54% in this study compared to Greater Houston at 16.6% is due to comfort with answering food insecurity screening in this setting might be an overstatement. People that attend this clinic were on Medicaid or Medicare, suggesting a higher probability of lower/fixed income.

Additionally, the method used to measure food insecurity in this study was the Hunger Vital Sign and the full USDA questionnaire may have been the method used to get the Greater Houston prevalence of food insecurity. Authors should report if this may have contributed to a difference in measurement for comparison.

• This is an excellent point about Medicaid and Medicare patients being on fixed incomes, I removed this section from the paper. As far as food insecurity rates in our sample versus Greater Houston, the 16.6% estimated food insecurity rate in Greater Houston comes from Feeding America’s food insecurity calculator, which estimates area-level food insecurity based on poverty rates and unemployment rates. Either way, I removed that section from the paper to avoid confusion.

The title for Table 2 does not make sense and appears to be copied from a different study.

• Thank you for this comment. I corrected the title of the table, added median values, test statistics, and added a footnote indicating significance level.

---

## [Decision Letter · Decision Letter 1]

4 Dec 2022

PONE-D-22-11199R1Examining Neighborhood-Level Hot and Cold Spots of Food Insecurity in Relation to Social Vulnerability in Houston, TexasPLOS ONE

Dear Dr. Ramphul,

Thank you for submitting your manuscript to PLOS ONE. After careful consideration, we feel that it has merit but does not fully meet PLOS ONE’s publication criteria as it currently stands. Therefore, we invite you to submit a revised version of the manuscript that addresses the points raised during the review process. Here are the comments from the reviewers that you should focus on:

The second paragraph indicates "a plethora of research..." and only one reference is cited in the text. The authors should revise this phrase to "A literature review..." instead of "A plethora of research.."

Please, revise all references cited in the text. When there are more than two references, the right way of citing is (3-6), not (3,4,5,6).

You should add that the study does not consider the different levels of Food insecurity as a limitation.

When reporting p values if the p =0.000 please indicate it as p< 0.001. In general p values smaller than 0.001 should be reported as p<0.001

The first paragraphs in material section and method section should come under the subheading "Data", as these two paragraphs explain the study sample or the data used.

You could have another subheading "Food insecurity" for the paragraphs that explain the tool used for measuring food insecurity.

All paragraphs explaining the statistical analysis conducted should be under the subheading "Statistical Analysis”.

We look forward to receiving your revised manuscript.

Kind regards,

Jim P Stimpson, PhD

Academic Editor

PLOS ONE

Journal Requirements:

Reviewers' comments:

Reviewer's Responses to Questions

**Comments to the Author**

1. If the authors have adequately addressed your comments raised in a previous round of review and you feel that this manuscript is now acceptable for publication, you may indicate that here to bypass the “Comments to the Author” section, enter your conflict of interest statement in the “Confidential to Editor” section, and submit your "Accept" recommendation.

Reviewer #1: All comments have been addressed

Reviewer #2: All comments have been addressed

Reviewer #3: (No Response)

2. Is the manuscript technically sound, and do the data support the conclusions?

Reviewer #1: Yes

Reviewer #2: (No Response)

Reviewer #3: Yes

3. Has the statistical analysis been performed appropriately and rigorously? 

Reviewer #1: Yes

Reviewer #2: Yes

Reviewer #3: Yes

4. Have the authors made all data underlying the findings in their manuscript fully available?

Reviewer #1: Yes

Reviewer #2: Yes

Reviewer #3: No

5. Is the manuscript presented in an intelligible fashion and written in standard English?

Reviewer #1: Yes

Reviewer #2: Yes

Reviewer #3: Yes

6. Review Comments to the Author

Reviewer #1: Thank you very much for the responses. They have provided sufficient revisions. this can be accepted in its present form

Reviewer #2: Thank you for allowing me to review this new version of the manuscript titled "Examining Neighborhood-Level Hot and Cold Spots of Food Insecurity in Relation to Social Vulnerability in Houston, Texas." All my comments have been addressed. However, some minor aspects still do not allow me to approve this new version, mainly in the introduction section.

The second paragraph indicates "a plethora of research..." and only one reference is cited in the text. I reviewed the answer to another reviewer's comment about this point. Even when the cited reference is a literature review, its use needs to be more coherent with the phrase at the beginning of the paragraph. The authors should use the original references or indicate "A literature review..." instead of "A plethora of research.."

Please, revise all references cited in the text. When there are more than two references, the right way of citing is (3-6), not (3,4,5,6).

Another limitation is that the study does not consider the different levels of Food insecurity. I suggest adding it.

Reviewer #3: The authors have addressed most of the comments provided in the previous review. However, I have two minor comments that could improve this paper before publishing.

1. When reporting p values if the p =0.000 please indicate it as p< 0.001. In general p values smaller than 0.001 should be reported as p<0.001

2. I think you can rearrange the materials and method section to make if flow better and easier to follow.

e.g. The first paragraphs in material section and method section should come under the subheading "Data", as these two paragraphs explain the study sample or the data used.

You could have another subheading "Food insecurity" for the paragraphs that explain the tool used for measuring food insecurity.

All paragraphs explaining the statistical analysis conducted should be under the subheading "Statistical Analysis"

7. PLOS authors have the option to publish the peer review history of their article (what does this mean?). If published, this will include your full peer review and any attached files.

Reviewer #1: **Yes: **farooq

Reviewer #2: No

Reviewer #3: No

---

## [Author Response · Author response to Decision Letter 1]

3 Jan 2023

Dear Reviewers:

Thank you for your comments on my manuscript entitled “Examining Neighborhood-Level Hot and Cold Spots of Food Insecurity in Relation to Social Vulnerability in Houston, Texas.” I edited the manuscript to address each point in the manner outlined below.

Sincerely,

Ryan Ramphul, PhD, MS

Reviewer #1: Thank you very much for the responses. They have provided sufficient revisions. this can be accepted in its present form

Reviewer #2: Thank you for allowing me to review this new version of the manuscript titled "Examining Neighborhood-Level Hot and Cold Spots of Food Insecurity in Relation to Social Vulnerability in Houston, Texas." All my comments have been addressed. However, some minor aspects still do not allow me to approve this new version, mainly in the introduction section. The second paragraph indicates "a plethora of research..." and only one reference is cited in the text. I reviewed the answer to another reviewer's comment about this point. Even when the cited reference is a literature review, its use needs to be more coherent with the phrase at the beginning of the paragraph. The authors should use the original references or indicate "A literature review..." instead of "A plethora of research."

• Thank you for pointing this out. I removed the phrase “A plethora of research” and clarified that this information came from a systematic review of literature on the impact of food insecurity on health outcomes.

Please, revise all references cited in the text. When there are more than two references, the right way of citing is (3-6), not (3,4,5,6).

• Thank you for this comment. I revised all references cited in the text. When there are more than two references, I cited as (3-6), not (3,4,5,6).

Another limitation is that the study does not consider the different levels of Food insecurity. I suggest adding it.

• Thank you, I added this point to my limitations paragraph.

Reviewer #3: The authors have addressed most of the comments provided in the previous review. However, I have two minor comments that could improve this paper before publishing. 

1. When reporting p values if the p =0.000 please indicate it as p< 0.001. In general p values smaller than 0.001 should be reported as p<0.001

• Thank you for catching this, I changed to report all p values smaller than 0.001 as p<0.001.

2. I think you can rearrange the materials and method section to make if flow better and easier to follow. e.g. The first paragraphs in material section and method section should come under the subheading "Data", as these two paragraphs explain the study sample or the data used. You could have another subheading "Food insecurity" for the paragraphs that explain the tool used for measuring food insecurity. All paragraphs explaining the statistical analysis conducted should be under the subheading "Statistical Analysis"

• Thank you for this comment, I rearranged the materials and methods section the way you suggested, and it does indeed flow better.

---

## [Editor Report · Decision Letter 2]

5 Jan 2023

Examining Neighborhood-Level Hot and Cold Spots of Food Insecurity in Relation to Social Vulnerability in Houston, Texas

PONE-D-22-11199R2

Dear Dr. Ramphul,

We’re pleased to inform you that your manuscript has been judged scientifically suitable for publication and will be formally accepted for publication once it meets all outstanding technical requirements.

Kind regards,

Jim P Stimpson, PhD

Academic Editor

PLOS ONE

---

## [Editor Report · Acceptance letter]

3 Mar 2023

PONE-D-22-11199R2 

Examining Neighborhood-Level Hot and Cold Spots of Food Insecurity in Relation to Social Vulnerability in Houston, Texas 

Dear Dr. Ramphul:

I'm pleased to inform you that your manuscript has been deemed suitable for publication in PLOS ONE. Congratulations! Your manuscript is now with our production department. 

Kind regards, 

on behalf of

Prof Jim P Stimpson 

Academic Editor

PLOS ONE